# Long-range, non-local switching of spin textures in a frustrated antiferromagnet

Shannon C. Haley [1,2] ✉, Eran Maniv[3], Shan Wu[1,2], Tessa Cookmeyer[1,2], Susana Torres-Londono[1], Meera Aravinth[1], Nikola Maksimovic[1,2], Joel Moore [1,2], Robert J. Birgeneau [1] & James G. Analytis [1,2,4] ✉

Antiferromagnetic spintronics is an emerging area of quantum technologies that leverage the coupling between spin and orbital degrees of freedom in exotic materials. Spin-orbit interactions allow spin or angular momentum to be injected via electrical stimuli to manipulate the spin texture of a material, enabling the storage of information and energy. In general, the physical process is intrinsically local: spin is carried by an electrical current, imparted into the magnetic system, and the spin texture will then rotate in the region of current flow. In this study, we show that spin information can be transported and stored "non-locally" in the material $Fe_xNbS_2$. We propose that collective modes can manipulate the spin texture away from the flowing current, an effect amplified by strong magnetoelastic coupling of the ordered state. This suggests a novel way to store and transport spin information in strongly spin-orbit coupled magnetic systems.

The semiconductor devices behind modern computers are rapidly approaching the physical limits of charge-based electronics, spurring research into novel materials that can enable 'spintronic' technologies that leverage the spin as well as the charge of an electron. Magnonics is an emerging subfield whereby the collective excitations of the magnetically ordered system, known as magnons, can be electrically stimulated[1]. Such materials have unique advantages because the length scale over which spin is coherently transported without loss can be very large, in contrast to flowing electrons whose spin decay is generally shorter[2]. In addition to spin coherence, there is also the challenge of spin-based memory. It has been shown that some antiferromagnetic (AFM) materials can store spin information through the electrical manipulation of AFM domains, although such technologies are thought to use spin-polarized electrical currents that, on general grounds, are intrinsically local in nature[3,4]. However, the role of strain in these phenomena has risen in prominence recently, as the electrical manipulation of certain insulating antiferromagnets has been shown to be driven by a combination of strain and thermal effects[5,6]—and with it, there is a growing understanding of the

possibilities for switching broader classes of antiferromagnets and for longer-range manipulations.

The compound $Fe_xNbS_2$ is an easy-axis antiferromagnet on a triangular sublattice that has been found to switch between distinct resistance states upon the application of DC current pulses along perpendicular directions[7]. Importantly, it appears that collective dynamics of the magnetic spin texture play an important role in this directional switching, with very high tunability by compositional changes about $x \simeq 1/3$[8]. This switching behavior was originally ascribed to a 90° reorientation of the in-plane component of the Néel vector, imparted via spin–orbit torque from an induced spin-polarization due to the Rashba–Edelstein Effect, which is allowed because of the broken inversion symmetry of the crystal lattice[4,7,9,10]. Further study found that the single-ion anisotropy in this system is strong enough to preclude the possibility of any significant in-plane moment, and so differing orientations of the Néel vector alone cannot be responsible for the anisotropic resistance we observe[11]. Instead, recent work has shown that there are two nematic and nearly degenerate antiferromagnetic ground states in $Fe_xNbS_2$, one in which aligned spins form stripes and one in which they form zig-zags[12]. The current pulse appears to rotate

[1]Department of Physics, University of California, Berkeley, CA 94720, USA. [2]Materials Sciences Division, Lawrence Berkeley National Laboratory, Berkeley, CA 94720, USA. [3]Department of Physics, Ben-Gurion University of the Negev, Beer-Sheva 84105, Israel. [4]CIFAR Quantum Materials, CIFAR, Toronto, ON M5G 1M1, Canada. ✉e-mail: shannon_haley@berkeley.edu; analytis@berkeley.edu

the principal nematic axis of the magnetic order. These orders compete, with the stripe order dominating at dilute compositions $x < 1/3$, and zig-zag at excess Fe composition $x > 1/3$[12]. According to a recent DFT study, which explores the Fermi surface anisotropies of the domains of the respective phases, this likely explains the opposite switching responses in identical device geometries, as shown in Fig. 1a and b[13]. Consider the domain structures in Fig. 1d–f. A given direction of switching pulse destabilizes domains whose principal axes are parallel to the applied current so that a pulse in the [100] direction will strongly disfavor one specific stripy domain and one specific zig-zag domain[13,14]. With respect to the principal axes, the conductivity tensor components $\sigma_{xx} > \sigma_{yy}$ for stripe domains and $\sigma_{xx} < \sigma_{yy}$ for zigzag domains, so when the current is applied along 45°, there are opposite switching responses in the off-diagonal resistivity[13]. At compositions where the order parameters are comparable in magnitude, one would expect the response to vanish—and this is exactly what is observed at $x = 0.33$, where the amplitude of the switching response is suppressed and a change in the sign of the response is observed as a function of the pulse current density[8]. A schematic of this mechanism is presented in Supplementary Fig. S13.

In this study, we show that a class of switchable, metallic antiferromagnets $Fe_xNbS_2$, exhibits the ability to manipulate spin information 'non-locally'—namely, tens of microns away from the electrical stimulus. This is orders of magnitude further than the spin diffusion length of typical metallic antiferromagnets[2,15–18]. We propose a picture that leverages two long-range effects: collective excitations to carry spin and strong magnetoelastic coupling to allow complex domain structures to propagate over large distances.

## Results

The two order parameters are known to have strong magneto-elastic coupling[19] and so it is likely that strain can be used to tune the switching behavior. To demonstrate this, we study the switching behavior under strain. Figure 1b, c show switching responses observed for the same $x = 0.31$ device, where Fig. 1b is the response after the device is cooled with no applied strain and Fig. 1c is the

response after being cooled with strain (corresponding to an applied 40 V to the piezoelectric cube the device was mounted on). At the current density shown, there is a change in the sign of the switching response due to the applied strain. At higher current densities, the original sign is recovered (see Fig. S1 in the supplement), so that there is a sign flip as a function of the pulse current density. This similarity to the $x = 0.35$ sample behavior could be explained by the strain subtly altering the RKKY-dominated exchange constants and allowing a slight increase in the minority zigzag phase, as illustrated in Fig. 1e. Supporting this interpretation, we note the difference in the lattice parameter between the zig-zag and stripe phase is comparable to the strain applied by the piezoelectric (see Supplementary Figs. S3–S6 for high-resolution cryogenic XRD). Strain can therefore tune the switching response for a $x = 0.31$ device, to that of a device with $x = 0.35$–a direct indication magneto-elastic coupling can be used to manipulate the domain structure of the magnetic texture.

With the importance of magneto-elastic coupling established, we now turn to how this can be leveraged to allow long-range transport of spin to store information. Heat capacity and magnetization measurements of characteristic samples are shown in Fig. 2a, b, respectively, showing magnetic transitions and spin glass behavior consistent with our previous characterizations of $Fe_{1/3}NbS_2$ and $Fe_{0.35}NbS_2$[11,20]. In Fig. 2c we illustrate a device designed to measure the non-local switching response of the antiferromagnetic texture of $Fe_xNbS_2$. DC current pulses are applied along the directions denoted as A and B, with a view to triggering magnons that can transport spin down the neck of the device. After the application of a pulse, the transverse resistance is measured with a probe current $I_{probe}$ at three distinct locations and goes to either a higher or lower resistance state, when a pulse current $I_{pulse}$ is applied in either the A or B direction. The low-temperature longitudinal resistivities of the devices measured had some small variations but were generally close to $10^{-4}$ Ω cm. The contacts marked $V_0$, which intersect the current pulse bars, will be referred to as local, and the contacts marked $V_1$ and $V_2$ will be referred to as non-local in this paper.

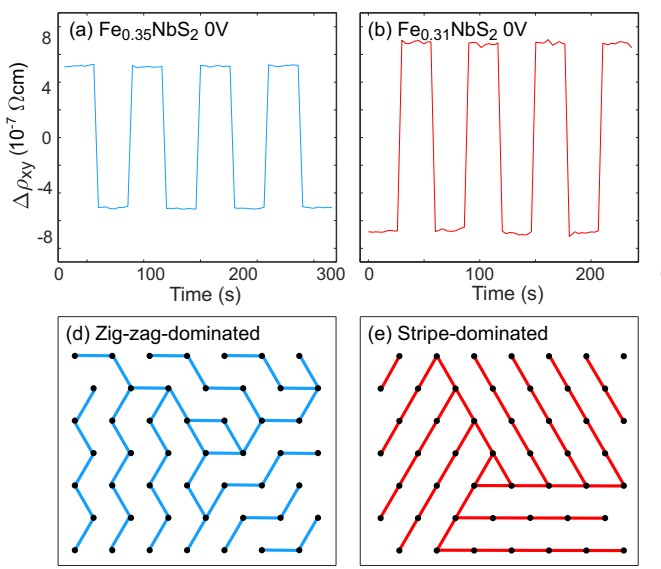

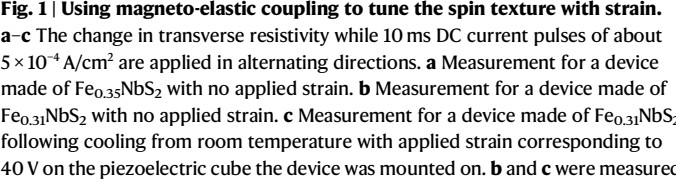

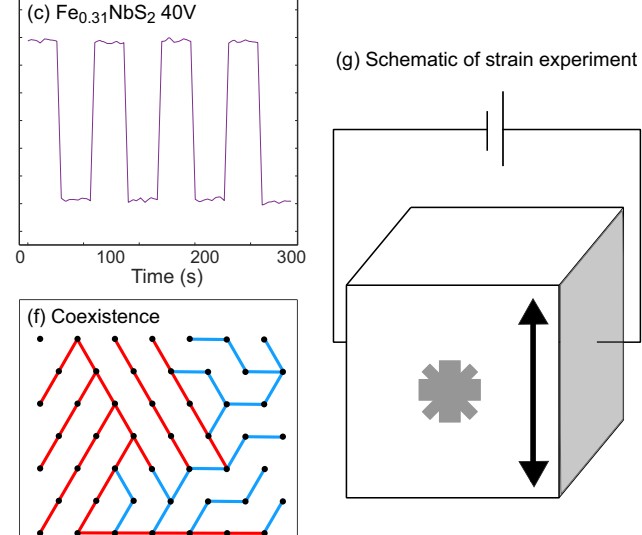

**Fig. 1 | Using magneto-elastic coupling to tune the spin texture with strain.**
**a–c** The change in transverse resistivity while 10 ms DC current pulses of about $5 \times 10^{-4}$ A/cm² are applied in alternating directions. **a** Measurement for a device made of $Fe_{0.35}NbS_2$ with no applied strain. **b** Measurement for a device made of $Fe_{0.31}NbS_2$ with no applied strain. **c** Measurement for a device made of $Fe_{0.31}NbS_2$ following cooling from room temperature with applied strain corresponding to 40 V on the piezoelectric cube the device was mounted on. **b** and **c** were measured

on the same device. Complete dataset with pulse current dependence is available in the supplement (Fig. S1). **d** Dominant spin texture in $Fe_{0.35}NbS_2$. **e** Dominant spin texture in $Fe_{0.31}NbS_2$. **f** Proposed spin texture in $Fe_{0.31}NbS_2$ cooled under strain. The proportion of zigzag phase is exaggerated. **g** Schematic of strain measurement. Voltage is applied between two electrodes around a cube of piezoelectric material, causing a directional expansion of the material which strains the device mounted on the cube.

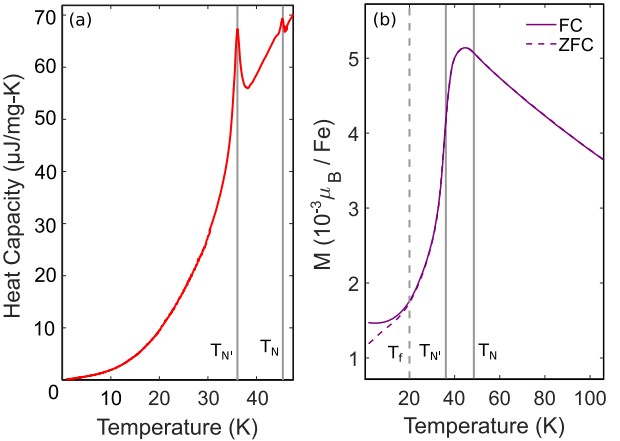

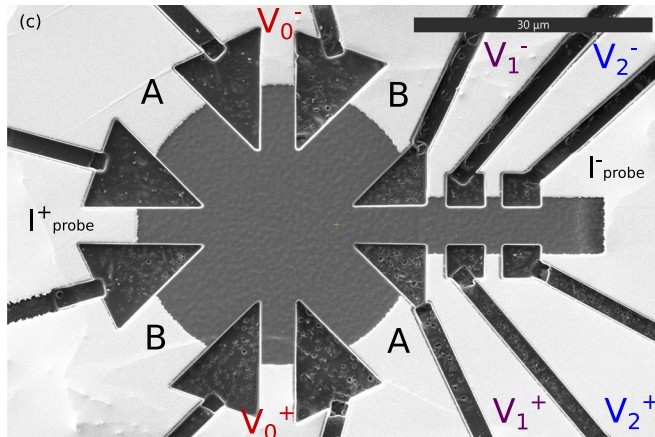

**Fig. 2 | Basic characterization of devices made of Fe$_{0.35}$NbS$_2$. a** Heat capacity as a function of temperature. Vertical solid lines marks $T_N$ and $T_{N'}$, the AFM transitions. **b** Magnetization as a function of temperature measured in 1000 Oe along the $c$-axis. The field-cooled (FC) measurement, shown as a solid curve, was measured from low to high temperature after cooling the sample in an 1000 Oe field. The zero-field-cooled (ZFC) measurement, shown as a dotted curve, was measured from low to high temperature after cooling the sample with no external field. Vertical dotted and solid lines indicate the onset of the spin glass behavior ($T_f$) and the AFM transitions ($T_N$ and $T_{N'}$), respectively. **c** A switching device made from a bulk crystal. The two pulse bars are marked $A$ and $B$. The AC probe current is applied along the path marked $I_{probe}$. The local signal is measured using the contacts labeled $V_0$, and the non-local signals are measured using the contacts labeled $V_1$ and $V_2$.

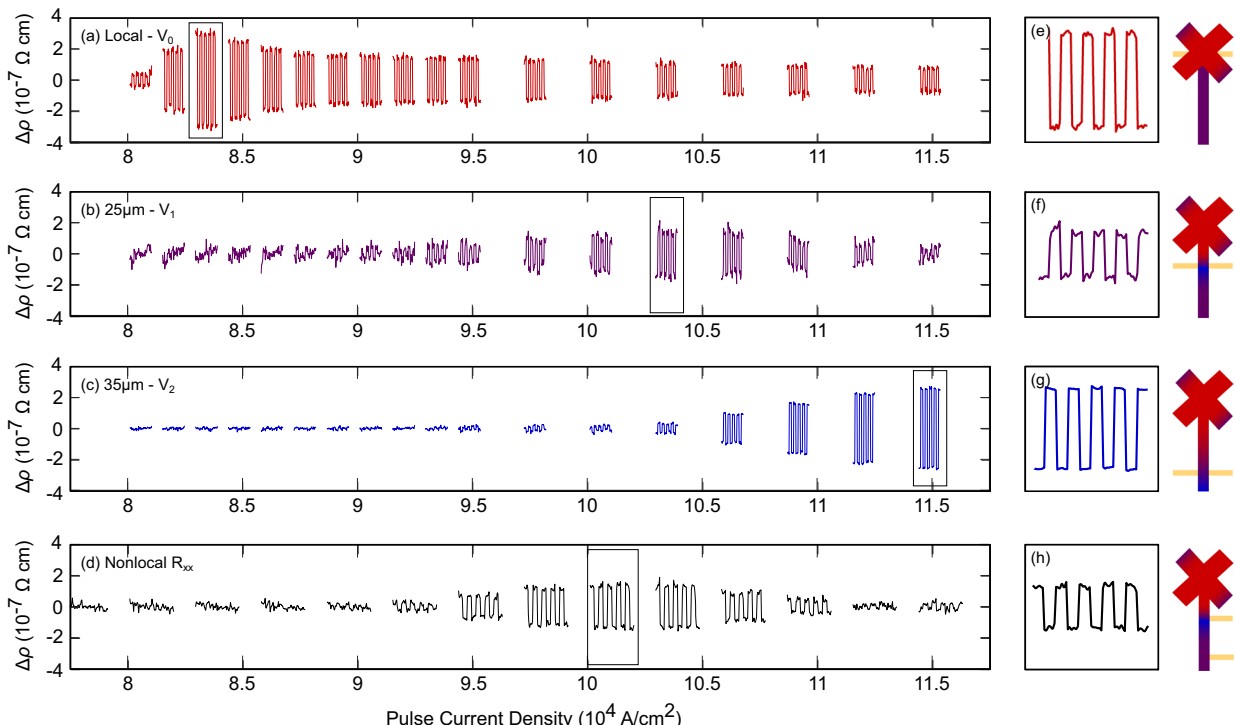

**Fig. 3 | Local and non-local switching of the antiferromagnetic spin texture.**
**a** Transverse resistance response measured locally between the contacts labeled $V_0$. **b** Transverse resistance response measured 25 μm from the center of the device, between the contacts labeled $V_1$. **c** Transverse resistance response measured 35 μm from the center of the device, between the contacts labeled $V_2$. **d** Longitudinal resistance measured on the non-local portion of the device, between two adjacent contacts labeled $V_1$ and $V_2$. **e–h** Single sets of switching responses at the current densities indicated on the left. Schematics to the right illustrate locations of measurement contacts, with shading indicating possible domain distribution at the given current density (red and blue are perpendicular domains and purple indicates multi-domain regions).

In Fig. 3 we show that the device has both local and non-local responses to the current pulses. Figure 3a shows the local response as a function of pulse current density. The response is not monotonic, turning on at about $8 \times 10^4$ A/cm$^2$, quickly reaching a maximum, and then decreasing slightly to reach a stable level around $11.4 \times 10^4$ A/cm$^2$. The measurements taken at 25 and 35 μm from the center of the active portion of the device are shown in Fig. 3b, c, respectively. The measurement taken 35 μm from the center requires a larger current density to register a change from the pulses than is necessary at 25 μm from the center, and both require larger current densities than the local response. The relative sizes of the responses vary from device to device, but the current density required is largely unchanged. Similar devices made of Fe$_x$NbS$_2$ $x \approx 1/3$ show weak local switching, but no stable switching response at the non-local contacts (see

Supplementary Fig. S2). The progressively larger current densities required to observe a switching response further from the active area of the device are largely consistent with the propagation of magnons, which dissipate with distance. There are two notably surprising aspects to this result, however. First, the non-local contact $V_2$, while requiring a larger current density, tends to have a larger switching response than the non-local contact $V_1$, which is closer to the active area. Second, the non-local contacts generally exhibit an opposite switching response to the local contacts $V_0$, so that the pulse directed in the same direction (A or B) will raise the local transverse resistance but lower the non-local transverse resistance. This suggests that the preferred domain orientation upon a current pulse differs between the two regions.

By symmetry, the longitudinal resistivity should be weakly affected by the current pulses[7], leaving domain wall scattering as the most likely origin of changes to $\rho_{xx}$. The response of $\rho_{xx}$ between the 25 and 35 μm non-local contact is shown in Fig. 3d. The non-local $\rho_{xx}$ response closely mimics the 25 μm non-local $\rho_{xy}$ response, with a peak just below $10.5 \times 10^4$ A/cm². The response of $\rho_{xx}$ is notably absent where the 35 μm non-local $\rho_{xy}$ response $V_2$ is strongest; this immediately suggests that there is significant domain wall scattering when $V_1$ has a maximal response and far less when $V_2$ has a maximum response. The opposite response between the local and 35 μm non-local switching not only suggests that the average principal axis of highest conductivity must be rotated in the perpendicular direction, but that there must also be fewer domains when this rotation occurs.

Figure 4b shows the temperature dependence of the pulse current with the maximum switching response for both local and non-local contacts (see Supplementary Figs. S7–S9). These measurements were taken on a device with non-local contacts 20 and 27 μm away from the center of the active area. At all three locations on the device, the threshold switching current grows with increasing temperature below the AFM temperature, closely mimicking neutron scattering measurements of the peak intensity corresponding to the AFM order parameter (Fig. 4a). This, and the disappearance of the switching response at the Néel temperature demonstrates a direct connection between the threshold current for switching and the opening of an AFM gap.

## Discussion

The temperature dependence of the switching amplitude shown in Fig. 4 is strongly indicative that the switching threshold current is proportional to the magnitude of the antiferromagnetic order parameter. The non-monotonic shape of the switching behavior as a function of current density observed locally is also observed in the non-local contacts, suggesting the same underlying behavior is also present in these regions.

Two unusual features from the data deserve some attention. (i) The non-local response closer to the active area ($V_1$) has a consistently smaller signal than that further away ($V_2$). (ii) The second non-local region has an average principal axis of the highest conductivity that is always oriented perpendicular to that of the local region.

We suggest that both of these effects are connected by the magneto-elastic response of the system. Little et al. recently showed that the antiferromagnetic order is strongly coupled to a structural distortion. Here, we have demonstrated that strain can directly control the sign of the switching response (Fig. 1b, c)[21]. The devices studied are clamped by the adhesive holding them to our test platform, so when the system undergoes the AFM transition, the minimization of the overall strain will favor a multi-domain state. This is similar to magnetostrictive effects in martensites which also balance elastic energy against the penalty of forming domain boundaries[22]. When the present system is switched in one region, the competition between bulk and surface strain produces a long-range field[22–25], rotating domains in the opposite direction to preserve the balance of different orientations.

Assuming the clamped boundary is the originator of the long-range elastic forces, it is natural to expect that this effect is most stark close to the crystal's edge, explaining the larger response in $V_2$ than $V_1$ because it is measured closer to the boundary. The 25 μm non-local response would then detect domain wall scattering and smaller re-orientations of the ordering vector, explaining its relatively smaller response in $\rho_{xy}$ and larger response in $\rho_{xx}$, which is amplified by domain boundary scattering. Finally, we note that in order for this mechanism to be effective, the internal strain of the device must be significant—comparable to the strains applied in our experiment

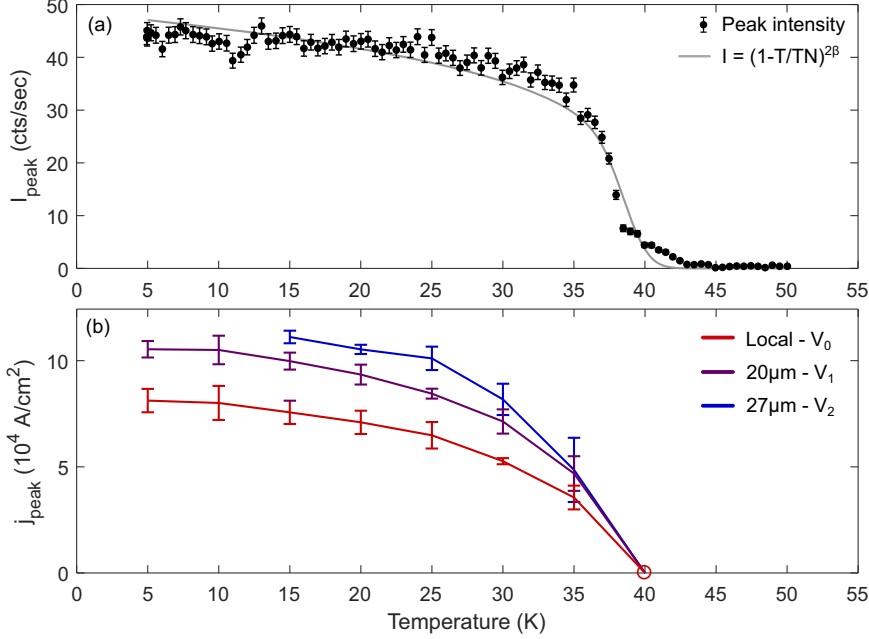

**Fig. 4 | Correspondence of threshold current and the antiferromagnetic order parameter. a** Order parameter peak intensity measured with neutron scattering as a function of temperature, with the associated critical exponent fit, with $2\beta = 0.21(2)$. Fit includes an assumed Gaussian distribution of N'eel temperatures with $\sigma = 1.9$ K. **b** Current density of peak switching response as a function of temperature, for all three sets of measurement contacts. Peak switching was determined by fitting the amplitudes of the responses to a Gaussian model, whose standard deviations give the uncertainty indicated by the error bars. The open circle at 40 K denotes the lack of switching at this temperature regardless of current density. The full dataset can be found in Supplementary Figs. S7–S9.

shown in Fig. 1g of ~0.1%. Given that anisotropy of the lattice parameters themselves is only ~0.1%, this suggests that the applied current pulses must orient a significant fraction of the device into a single domain, away from the active area. This also explains the reduction of $\rho_{xx}$ at higher currents, as a consequence of reduced domain walls.

Disorder[26,27], entropy[28,29], and leakage current could provide alternative explanations for the equilibrium domain configuration between pulses. Disorder-driven domain formation, however—in which domains are tied to defects—does not explain the stronger signal at the farther non-local contact nor its sign being opposite to the local contacts. Similarly, leakage current also does not explain why switching is so much stronger in the farther non-local contact $V_2$ than in $V_1$. Entropy-driven domain formation should be strongest close to the Néel temperature, which is inconsistent with the switching being enhanced as the temperature is lowered below the transition. The elastic picture by contrast is very natural in this system and, given the known structural anisotropy[19], must be active.

However, the elastic response does not alone explain why the current pulses can orient domains so far from the region where the current actually flows. Typically, metals transport spin via their conduction electrons, while magnetic insulators transport spin through collective excitations such as magnons. Conduction electron spin currents generally decay more quickly than magnon spin currents, and in practice, antiferromagnetic metals, in particular, tend to have very short spin diffusion lengths, largely around or under 2 nm—as is, for example, the case in Mn-based alloys[2,30–33]. In magnetic insulators, on the other hand, spin decay in single-crystal systems has been extended to ten microns (see the case of $\alpha$-Fe$_2$O$_3$[18]). To account for the long distances of spin transport observed here, the transport medium in the present system is likely to also leverage collective modes. Our data suggest that the combined action of spin-carrying collective excitations and the magnetoelastic coupling of the system allow regions of the sample that are tens of microns away to be switched. These scales are orders of magnitude larger than spin decay lengths of typical metallic antiferromagnets, which is a welcome discovery relevant for potential technologies based on such materials[2]. One question is to which order the collective excitations belong. A natural candidate is the antiferromagnetic order itself, whose magnons transfer their spin to the nearest domain wall, but there could also be collective dynamics associated with the competition of the coexisting zig–zag and stripe order parameters. Future work is needed to reveal the precise nature of these ingredients and to elucidate the roles that they play in order to form a full microscopic picture of the mechanism behind the non-local response. For now, it would be interesting to see whether other electrically switchable antiferromagnets can show similar behavior.

## Methods

Single crystals of Fe$_x$NbS$_2$ were synthesized using a chemical vapor transport technique. A polycrystalline precursor was prepared from iron, niobium, and sulfur in the ratio $x$:1:2 (Fe:Nb:S). The resulting polycrystalline product was then placed in an evacuated quartz ampoule with iodine as a transport agent (2.2 mg/cm$^3$), and put in the hot end of a two-zone MTI furnace with temperature set points of 800 and 950 for a period of 7 days. High-quality hexagonal crystals with diameters up to several millimeters were obtained. The crystal structure was checked by powder diffraction (see Supplementary Figs. S3–S6).

Devices were fabricated using the FEI Helios G4 DualBeam focused ion beam at the Molecular Foundry at LBNL. The devices were mounted on Torr Seal and sputtered with gold for electrical contact. In most cases, the crystals were exfoliated to reach a thickness under 4 μm. The switching pulses were single square waves administered with Keithley 6221 Current Sources.

Transport was measured during the switching experiments via an MFLI lock-in amplifier. An AC probe current ran through the device

both during and in between switching events and each measurement in this work had an rms value between 25 and 100 μA and a frequency of either 277 or 1333 Hz. Measurements were also taken with the AC probe current turned off and its corresponding leads removed during the switching event itself, and the resulting switching behavior was unchanged. A range of AC probe frequencies was also tested, and aside from an increase or decrease in noise, there was no measurable difference in the resulting behavior. Both of these tests can be found in Supplementary Figs. S10–S12. Data from additional devices are presented in Fig. S14.

Low field magnetization measurements were performed using a Quantum Design MPMS-3 system with a maximum applied magnetic field of 7 T. AC heat capacity was measured using a Quantum Design PPMS system.

High-resolution wide-angle x-ray powder diffraction measurements were performed on the beamline 28-ID-1 at the National Synchrotron Light source II at Brookhaven National Laboratory. The raw data were collected by the incident beam with a wavelength of 0.1668 Å and a Perkin–Elmer area detector and transformed into diffraction data. The Rietveld refinement was carried on by GSAS-II[34]. Single-crystal neutron diffraction experiment was performed on BT-7 at the NIST Center for Neutron Research.

## Data availability

The raw data for analyses presented in this manuscript can be found on OSF (https://doi.org/10.17605/OSF.IO/RMD5K). Information and data from additional measurements are available upon request.

## Code availability

The code used to analyze the data in this work is available from the corresponding author on reasonable request.

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

## Acknowledgements

This work was supported as part of the Center for Novel Pathways to Quantum Coherence in Materials, an Energy Frontier Research Center funded by the U.S. Department of Energy, Office of Science, Basic Energy Sciences. Work by J.G.A. was partially supported by the Gordon and Betty Moore Foundation's EPiQS Initiative through Grant No. GBMF9067. The work by S.W. and R.J.B. was funded by the U.S. Department of Energy, Office of Science, Office of Basic Energy Sciences, Materials Sciences and Engineering Division under Contract No. DE-AC02-05-CH11231 within the Quantum Materials Program (KC2202).

## Author contributions

S.C.H., S.T.-L., and M.A. performed crystal synthesis. S.C.H. and E.M. performed Focused Ion Beam fabrication. S.C.H. conducted switching measurements. E.M. conducted magnetization measurements. N.M. and E.M. conducted heat capacity measurements. T.C. and J.M. discussed the consequences of different kinds of nonlocal magnetization switching. S.W. and R.J.B. conducted neutron scattering measurements and synchrotron x-ray powder diffraction and analyzed the resulting data. J.G.A. designed the main experiment. S.C.H. and J.G.A. performed data analysis. S.C.H., J.G.A., T.C., and J.M. wrote the manuscript with input from all co-authors.

## Competing interests

The authors J.G.A., S.H., and E.M. have submitted a patent application relevant to the contents of this manuscript. The remaining authors declare no competing interests.
