## [Peer Review File · Nature Communications]

Reviewers' Comments:

Reviewer #1:

Remarks to the Author:

Upon transfer of this manuscript from Nature to Nature Communications, the authors have made further revisions and clarifications to criticism from several referees. In my opinion, the authors have made a complete and faithful attempt to address all referees concerns. The result is an improved manuscript, and one that communicates fascinating and important results on a novel implementation of non-local effects in a new area of antiferromagnetic spintronics.

As the reports of the other referees indicated a clear willingness to publish, though not in Nature, and considering additional efforts the authors made to respond to all reports, I am very happy to recommend publication in Nature Communications.

Reviewer #2:

Remarks to the Author:

The authors present a set of measurements on Fe_xNbS₂, including multi-terminal transport properties with local and non-local electrical probes. The results are highly interpreted, and I found the explanation to be quite complex. It is therefore difficult to ascertain the reliability / veracity of the explanation. To be more specific with my concern, I list here some of the ingredients of the system description:

We have an AFM with two different order types, zig-zag and stripe. Each AFM order type has a 3-fold degeneracy with respect to orientation. We therefore have 6 domains. In addition, previous papers have invoked a spin glass phase co-existing with these domains to describe the current-induced torque effect on the domain structure (Sci. Adv. 2021; 7 : eabd8452). This is, by itself, an interesting but very complex and unexplored scenario. The resistivity depends on the relative orientation of the current and the average domain orientation (averaged over domain distribution). This effect is straightforward and expected. Finally, the strain in the system will also affect the domain distribution.

With this in mind, the authors present a transport study in which the resistivity changes locally and non-locally upon the application of current through a multi-terminal device. Non-local changes in the resistance are observed, and are the focus of the paper. These signals are related to the antiferromagnetic order, as they disappear above the Neel temperature. The remainder of the paper provides a proposed mechanism for the non-local change in measured resistance.

I find the description of the proposed mechanism to be unclear. Both strain and spin transport are mentioned as essential. However at different points, one or the other is described as the dominant mechanism, or that both are coupled together in some essential way. For example, it's stated that "long-range switching is driven by long-range strain" (response to referee 4), or "combined action of spin-carrying collective excitations coupled to the magnetoelasticity of the system". I just can't get a clear picture of the mechanism. Does the current induce magnetic excitations, which are transduced to strain which propagates through the material, which are then transduced back to magnetic excitations? When the excitation arrives at the nonlocal position, do we invoke the spin-glass related spin-orbit torque of Ref. (Sci. Adv. 2021; 7 : eabd8452) for domain wall motion? I find it quite difficult to follow at this point. While it's not necessary to have a full comprehensive account of all the microscopics of this very complex system for publishing this paper, I don't feel that the data presented enable definitive statements to be made, particularly in this rather complex scenario.

One point made by the authors in their responses to the other referees deserves clarification. The Rashba-Edelstein does **not** lead to spin-polarized currents, at least to linear order in the applied E-field. The Rashba-Edelstein effects leads to spin densities. Perhaps the authors have in mind that the current is then polarized by this non-equilibrium spin density. This is true, however both current and non-equilibrium spin density are linear order in E-field, so that the resulting spin-polarized current would be second order in E-field. On the other hand, the non-equilibrium spin density may diffuse to regions away from the current source. This diffusive spin current would, however, be subject to short spin lifetimes which the authors assume are present in this material.

Response to Referee 5

We would like to thank the reviewer for their constructive review. We have made several changes to the manuscript that clarify the points highlighted and made the manuscript significantly more concise and succinct.

The authors present a set of measurements on Fe_xNbS₂, including multi-terminal transport properties with local and non-local electrical probes. The results are highly interpreted, and I found the explanation to be quite complex. It is therefore difficult to ascertain the reliability / veracity of the explanation. To be more specific with my concern, I list here some of the ingredients of the system description:

We thank the referee for giving us the opportunity to clarify our arguments.

We have an AFM with two different order types, zig-zag and stripe. Each AFM order type has a 3-fold degeneracy with respect to orientation. We therefore have 6 domains. In addition, previous papers have invoked a spin glass phase co-existing with these domains to describe the current-induced torque effect on the domain structure (Sci. Adv. 2021; 7 : eabd8452). This is, by itself, an interesting but very complex and unexplored scenario. The resistivity depends on the relative orientation of the current and the average domain orientation (averaged over domain distribution). This effect is straightforward and expected. Finally, the strain in the system will also affect the domain distribution.

With this in mind, the authors present a transport study in which the resistivity changes locally and non-locally upon the application of current through a multi-terminal device. Non-local changes in the resistance are observed, and are the focus of the paper. These signals are related to the antiferromagnetic order, as they disappear above the Neel temperature. The remainder of the paper provides a proposed mechanism for the non-local change in measured resistance.

I find the description of the proposed mechanism to be unclear. Both strain and spin transport are mentioned as essential. However at different points, one or the other is described as the dominant mechanism, or that both are coupled together in some essential way. For example, it's stated that "long-range switching is driven by long-range strain" (response to referee 4), or "combined action of spin-carrying collective excitations coupled to the magnetoelasticity of the system". I just can't get a clear picture of the mechanism. Does the current induce magnetic excitations, which are transduced to strain which propagates through the material, which are then transduced back to magnetic excitations? When the excitation arrives at the nonlocal position, do we invoke the spin-glass related spin-orbit torque of Ref. (Sci. Adv. 2021; 7 : eabd8452) for domain wall motion? I find it quite difficult to follow at this point. While it's not necessary to have a full comprehensive account of all the microscopics of this very complex system for publishing this paper, I don't feel that the data presented enable definitive statements to be made, particularly in this rather complex scenario.

We agree with the reviewer that the arguments made were unclear. We argue that both magnetoelastic coupling and spin transport via collective modes are important. The former is on firmer ground – we have direct evidence of the structural and electronic anisotropy of the domains from prior work. Furthermore, we establish in the first part of the manuscript a direct connection between a strain field and the magnetic order. So there should be little doubt that magneto-elastic coupling is at play in leading to long range effects the device. The spin transport by collective modes is indirectly established by the data; the effect of the current pulses is seen in regions where the current is not present, and a significant fraction of the device must be in a single domain to create enough strain to observe the non-local effects. This evidence is consistent with spin transport via collective modes (or at least not mediated by

conduction electrons), but it is not direct evidence as the reviewer points out. We clarify this in the text as well as the reviewer's point below.

*One point made by the authors in their responses to the other referees deserves clarification. The Rashba-Edelstein does *not* lead to spin-polarized currents, at least to linear order in the applied E -field. The Rashba-Edelstein effects leads to spin densities. Perhaps the authors have in mind that the current is then polarized by this non-equilibrium spin density. This is true, however both current and non-equilibrium spin density are linear order in E -field, so that the resulting spin-polarized current would be second order in E -field. On the other hand, the non-equilibrium spin density may diffuse to regions away from the current source. This diffusive spin current would, however, be subject to short spin lifetimes which the authors assume are present in this material.*

The reviewer is correct, and we have emphasized that spin diffusion is another possibility of spin transport, though the spin lifetime and mean free path would have to be much longer than typical antiferromagnetic metals.